# Research on the internal flow and macroscopic characteristics of a diesel fuel injection process

**Hua Xia** [ORCID] *

School of Automotive and Traffic Engineering, Jiangsu University, Zhenjiang, Jiangsu, China

* 250930192@qq.com

**Data Availability Statement:** All relevant data are within the paper.

**Funding:** This work was supported by Postgraduate Research & Practice Innovation Program of Jiangsu Province (No.KYCX17_1777). The funders had no role in study design, data

## Abstract

The internal flow and macroscopic spray behaviors of a fuel injection process were studied with schlieren spray techniques and simulations. The injection pressures($P_{in}$)and ambient pressures($P_{out}$)were applied in a wide range. The results showed that increasing the $P_{in}$ is likely to decrease the flow performance of the nozzle. Furthermore, increasing the $P_{in}$ can increase the spray tip penetration. However, the effect of $P_{in}$ on the spray cone angle was not evident. The spray cone angle at an injection pressure of 160MPa was 21.7% greater than at a pressure of 100MPa during the initial spraying stage. Additionally, the discharge coefficient increased under high $P_{out}$, and the decrease in $P_{out}$ can promote the formation of cavitation. Finally, increasing the $P_{out}$ can decrease the penetration, while the spray angle becomes wider, especially at the initial spray stage, and high $P_{out}$ will enhance the interaction of the spray and the air, which can enhance the spray quality.

## Introduction

In diesel engines, the major emissions are soot and nitrogen oxide ($NO_x$), which are affected by spray quality [1–4]. Several studies have shown that improving the injection pressure can strongly enhance fuel atomization and spray quality [5–7]. Consequently, it is important to characterize the influences of injection pressure and back pressure on the inner flow in a diesel nozzle and the spray behaviors. In particular, diesel nozzles have small diameters and are subject to ultra-high injection pressures.

Agarwal et al. studied the influence of fuel injection pressure on spray behaviors with biodiesel in a constant volume chamber and found that the penetration was greater at higher injection pressures after the start of injection (ASOI) [8]. Zhu et al. studied the influences of injection pressure and ambient pressure on the mass flow rate with a laser-induced fluorescence technique, and their results indicated that increasing the injection pressure could increase the mass flow rate [9]. Moon et al. used an X-ray technique to investigate the effects of pressure on near-field spray dynamics, and their results indicated that a higher injection pressure could increase the spray velocity [10]. Eagle et al. examined macro spray behaviors under high injection pressures, and their results showed the fluctuating structures of multi hole

collection and analysis, decision to publish, or preparation of the manuscript.

**Competing interests:** The authors have declared that no competing interests exist.

sprays at high injection pressure during early on set [11]. Jia et al. examined the effects of injection pressure on the penetration length, and their results showed that a higher injection pressure could increase penetration, but the spray cone angle did not change much [12]. Kawaharada et al. examined the influence of injection pressure on the velocity and size of droplets in diesel spray. Their results showed that the droplet size under higher injection pressure was larger than the size under lower injection pressure at the spray center, but the reverse was true at the spray periphery [13]. Wang et al. studied the influence of injection pressure fluctuations on spray behaviors, they showed that injection pressure fluctuations can increase the mixture quality [14]. Payri et al. studied the influence of injection pressure fluctuations on spray behaviors and showed that injection pressure fluctuations could increase the mixture quality [15]. Payri et al. examined the effect of pressure on spray velocity through the particle image velocimetry (PIV) measurement method and found that back pressure could shorten the penetration and that injection pressure could increase the droplet velocity.

In addition, many experts and scholars have carried out numerous studies on the effects of ambient pressures on inner flows and spray behaviors. Manin et al. performed spraying a thigh back pressure and temperature through a high-speed long-distance microscopy experiment using n-dodecane and concluded that increasing the high back pressure can improve spray quality [16]. Sepret et al. experimentally studied the effect of ambient density on spray mixtures, and their results showed that ambient density can enhance gas entrainment intensity, thus improving the quality of diesel spray [17]. Li et al. studied diesel spray behavior under various injection pressures and back pressures, and the penetration and velocity were found to be different at the same back pressure when different back pressures were used because of gas compressibility [18]. Other researchers have also investigated the influence of back pressure on spray behaviors [19–21]. Their results have shown that back pressure has an impact on spray behavior.

This literature review shows that the influence of injection pressure and back pressure on spray behavior has been well studied. However, the flow dynamics mechanism and its effect on spray behavior at various pressures is still unclear because of the complexities of the cavitation flow in the nozzle orifice, such as when the cavitation phenomena and turbulence become more intensive at high injection pressures. These complexities motivate us to further our understanding of inner flows and spray behaviors at various injection pressures and ambient pressures.

The main purpose of this work is to examine internal flow character rizations using simulations and to study spray behaviors experimentally. First, we examine the impact of $P_{in}$ and $P_{out}$ on inner flows, including the cavitation generated inside the nozzle, and the flow parameters. Second, the spray behaviors (including the spray penetration, tip velocity, and spray cone angle) are assessed for different $P_{in}$ (100MPa, 120MPa, 140MPa, 160MPa) and $P_{out}$ (1MPa, 2MPa, 3MPa, 4MPa) through a spray test platform using the schlieren method. These studies provide critical insights by combining the internal flow and spray characteristics for various $P_{in}$ and $P_{out}$.

## The nozzle geometry and boundary conditions

### The nozzle geometry

A diesel nozzle with eight orifices was selected. Because of the symmetry of the diesel nozzle, only one of the eight orifices (45°) was calculated. The orifice of the cylindrical nozzle was used for the present investigation, as shown in Fig 1. The geometrical parameters of the orifice can be found in Table 1. The definition of the *K*-factor used in conical nozzles, and the

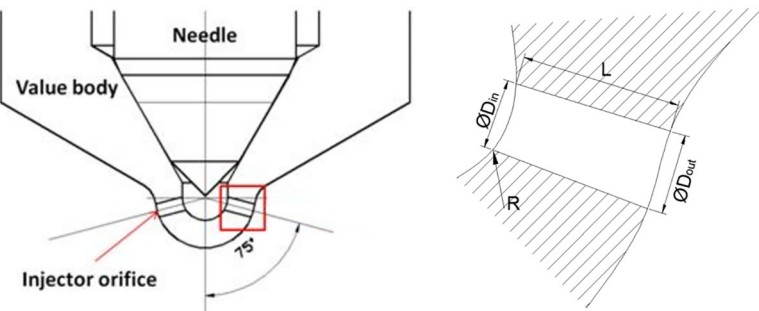

**Fig 1. Nozzle geometry parameter schematic diagram.**

definition of the *K*-factor are as follows [22]:

$$K = \frac{D_{in} - D_{out}}{L} \times 100 \qquad (1)$$

## The groups of operational conditions

To study the effect of $P_{in}$ and $P_{out}$ on the internal flow and spray behaviors, respectively, four groups of various $P_{in}$ and $P_{out}$ were selected for experimental and simulation research. The specific parameters are shown in Table 2.

## Description of the CFD method

A numerical method was used to study the inner flow of the nozzle. Fluent 14.0 was adopted to simulate the internal flow calculations. The mixture model was used to study the multiphase flow. The study was based on the pressure correction method and uses the SIMPLE algorithm [23]. The Schnerr and Sauer model was used as the cavitation model [24]. The turbulence model was the standard $k-\varepsilon$ model [25]. All simulations presented here were made using the standard $k-\varepsilon$ model with standard wall functions.

The meshing process of the geometries are presented in Fig 2. Because the mesh size has an impact on numerical results, the nozzle orifice inner flow was initially calculated with different mesh sizes [26]. Fig 3 shows that the $P_{in}$ was 120MPa and the back pressure was 5MPa. The mass flow rate of the orifice exit was plotted with various mesh sizes. The results showed that when the mesh size decreased from 30μm to 2μm, the mass flow rate tended to be consistent, and the mass flow rate was approximately 6.27 g/s with a grid size of 5μm. Consequently, with a comprehensive consideration of the calculation time and accuracy, a grid size of 5μm was used in this study, and the total number of grids was 0.63 million.

## Experimental setup and data processing

### Spray test platform

Fig 4 shows the EFS8400 spray test platform, which is adopted to capture spray images at different pressures and diameters. To obtain more distinct spray behaviors, the images are obtained by the schlieren method. The test platform includes a volume chamber, fuel injection

**Table 1. The geometrical characteristics of the orifices.**

| Orifice Type | $D_{in}$/mm | $D_{out}$/mm | $L$/mm | $R$/mm | $K$ |
|---|---|---|---|---|---|
| Cylindrical | 0.16 | 0.16 | 0.75 | 0 | 0 |

**Table 2. Operational conditions.**

| Case | $P_{in}$/MPa | $P_{out}$/MPa |
|---|---|---|
| 1 | 100 | 5 |
| 2 | 120 | 5 |
| 3 | 140 | 5 |
| 4 | 160 | 5 |
| 5 | 120 | 1 |
| 6 | 120 | 2 |
| 7 | 120 | 3 |
| 8 | 120 | 4 |

system, control unit and high-speed camera system. The nitrogen is stored in the gas cylinder, and the back pressure is controlled by a computer system. The maximum back pressure is 5.2MPa at room temperature. The fuel injection system is dominated by a high-pressure fuel pump, and a common rail injection system is applied to provide various $P_{in}$. The diesel injection duration time is approximately 1ms. Thus, during the schlieren experiment, the injection duration time is 1ms. In addition, standard diesel fuel is used during the spray test.

### Experimental data processing and error analyzing

Despite careful measurement preparations, four images are taken for each of the injection times, and the spray behaviors are averaged four times. The spray characteristics are thus taken as eight spray averages. The data processing is shown in Fig 5. When the spray cone angle is analyzed, the maximum error is located near the nozzle, where the spray cone angle (θ) is smaller, which causes a 4.6% maximum error in the spray cone angle. At the downstream of the spray, the maximum error for the spray cone angle is 1.8%. However, near the nozzle, where the concentration and hence contrast is high, this error should be systematic. To analyze the statistical errors, standard deviation and 96% confidence values are calculated for the spray tip penetration and the spray cone angle.

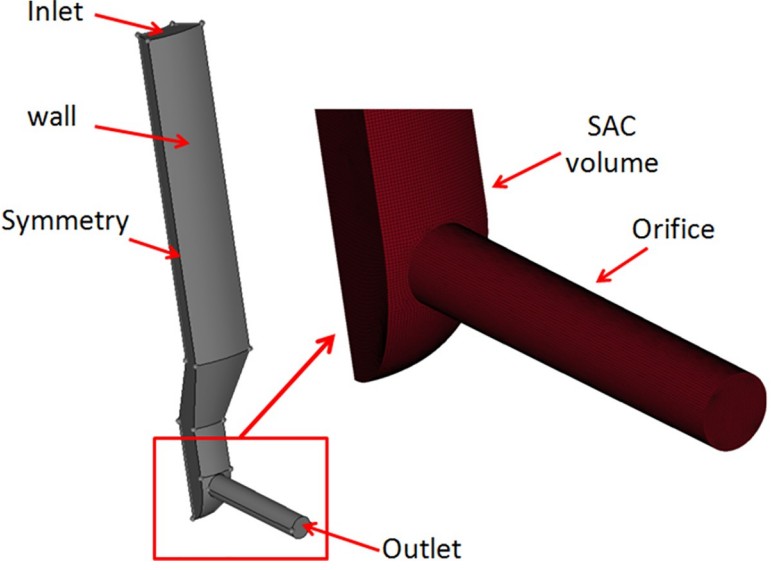

**Fig 2. Computational nozzle model and orifice mesh structure.**

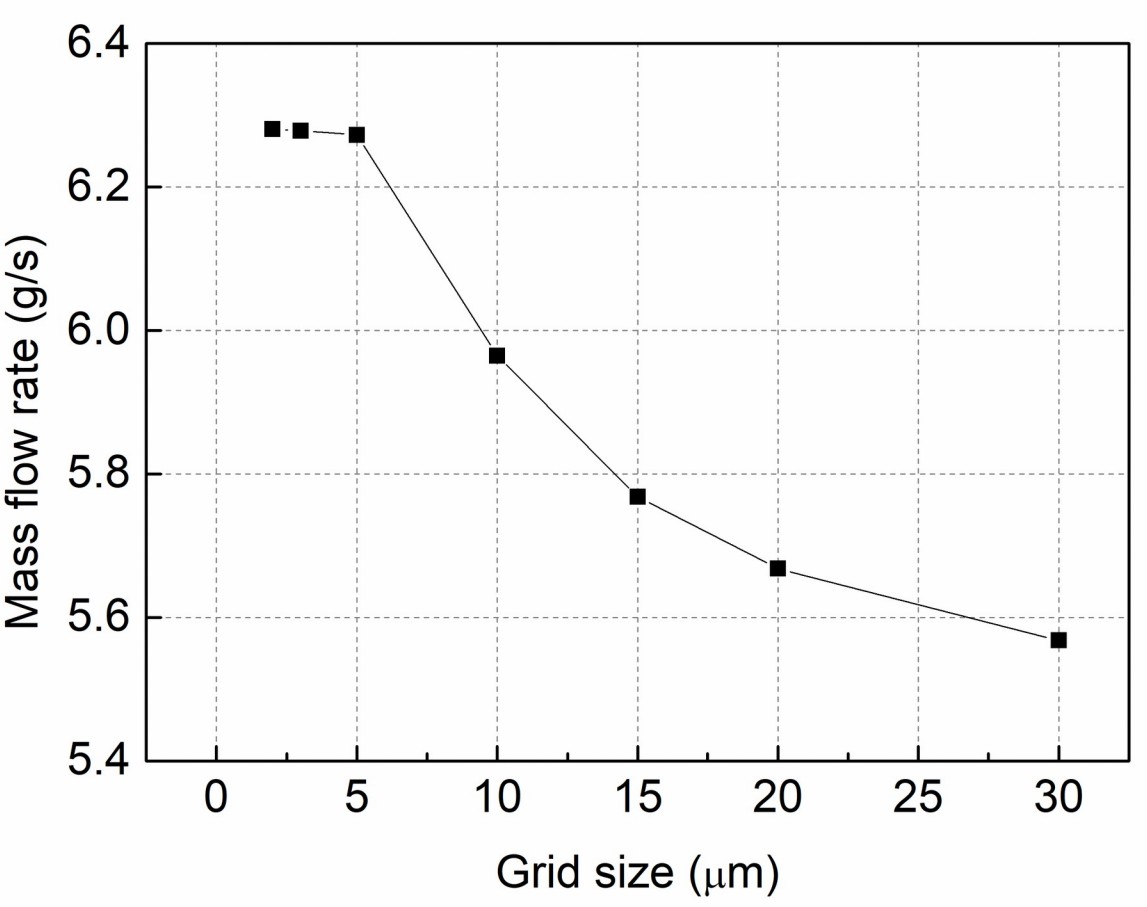

**Fig 3. Mass flow rates for different grid sizes.**

## Results and discussion

### Effects of $P_{in}$ on internal flow and spray characteristics

The purposes of the simulations and experimental tests presented here are to compare the inner flow and spray behaviors, with particular attention to the impacts of the injection

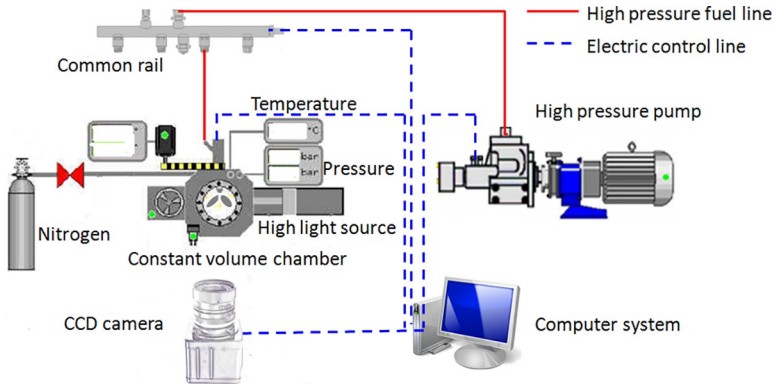

**Fig 4. Visualization spray test platform.**

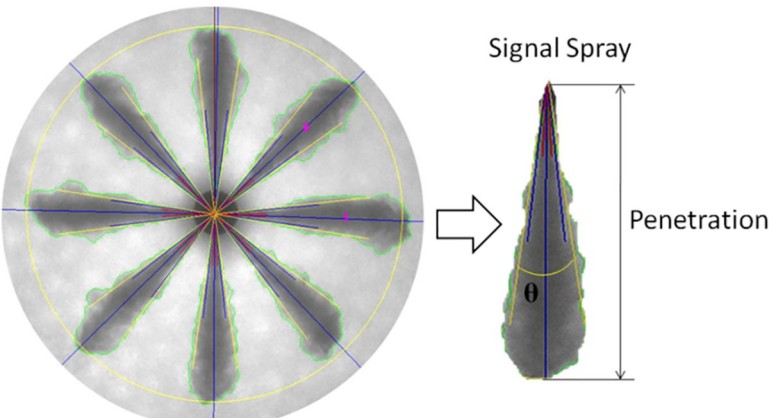

**Fig 5. Data processing.**

pressure on the inner flow and spray macroscopic parameters. Orifice exit flow parameters, vapor volume fractions and spray characteristics have been identified.

**Orifice exit flow parameters.** The discharge coefficient is calculated by combining the Bernoulli equation and the mass conservation equation:

$$C_d = \frac{\dot{m}}{A\sqrt{2\rho(P_{in} - P_{out})}} \tag{2}$$

where m $\dot{m}$ is the mass flow rate, $P_{in}$ is the injection pressure, $P_{out}$ is the ambient pressure, $\rho$ represents the liquid density, and $A$ is the geometric cross-section of the orifice.

Fig 6 represents the simulation results of mass flow rate and discharge coefficient at various injection pressures and a back pressure of 5MPa. Fig 6 shows that the mass flow rate is increased with increasing injection pressure. In particular, when the injection pressure increases from 100MPa to 160MPa, the exit mass flow rate increases by approximately 26.5%. In addition, the discharge coefficient decreases with increasing injection pressure. This means that the flow performance will be slightly decreased under the condition of high injection pressure, mainly due to the intensive cavitation disturbance in the nozzle with high injection pressure, which can suppress the flowing performance.

**Simulation results of vapor volume fraction distribution.** Fig 7 shows the contours of the volume fraction for the vapor phase at various $P_{in}$ (100MPa, 120MPa, 140MPa, 160MPa) and $P_{out}$ of 5MPa. For the vapor phase fraction contour, the zones colored in red illustrate where the cavitation is generated. The results show that the cavitation domain is generated near the inlet corner of the nozzle, and the cavitation becomes more intense with increasing injection pressures, which means that increasing the injection pressure can enhance the formation of cavitation. In addition, it can be seen in the reddomain inside the nozzle that the red domain extends to the nozzle exit at an injection pressure of 160MPa, but for an injection pressure of 100MPa, this phenomenon does not occur. This indicates that the cavitation flow distribution is extensive at high injection pressures. In addition, the cavitation bubbles break, which can increase the turbulence intensity in the diesel nozzle and thus decrease the flow performance of the nozzle. This is also the reason for the lowest discharge coefficient at an injection pressure of 160MPa.

**Experimental results of spray characteristics.** The spray morphology was obtained from the experimental spray method at different injection pressures. Fig 8 shows the evolution of the spray morphology at injection pressures of 100MPa, 120MPa, 140MPa, and 160MPa and

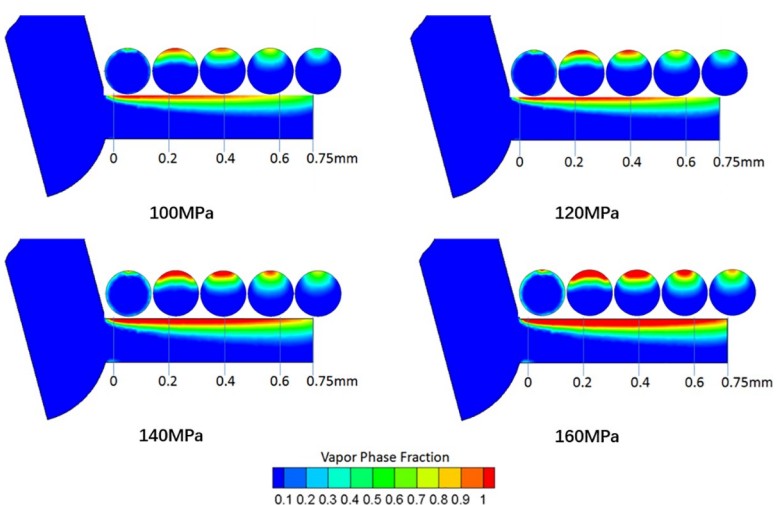

**Fig 6. Mass flow rate and discharge coefficient at various $P_{in}$ and fixed $P_{out}$ of 5MPa.**

100MPa

120MPa

140MPa

160MPa

Vapor Phase Fraction

0.1 0.2 0.3 0.4 0.5 0.6 0.7 0.8 0.9 1

**Fig 7. Contour of vapor volume fraction against different $P_{in}$ at various positions and fixed $P_{out}$ of 5MPa.**

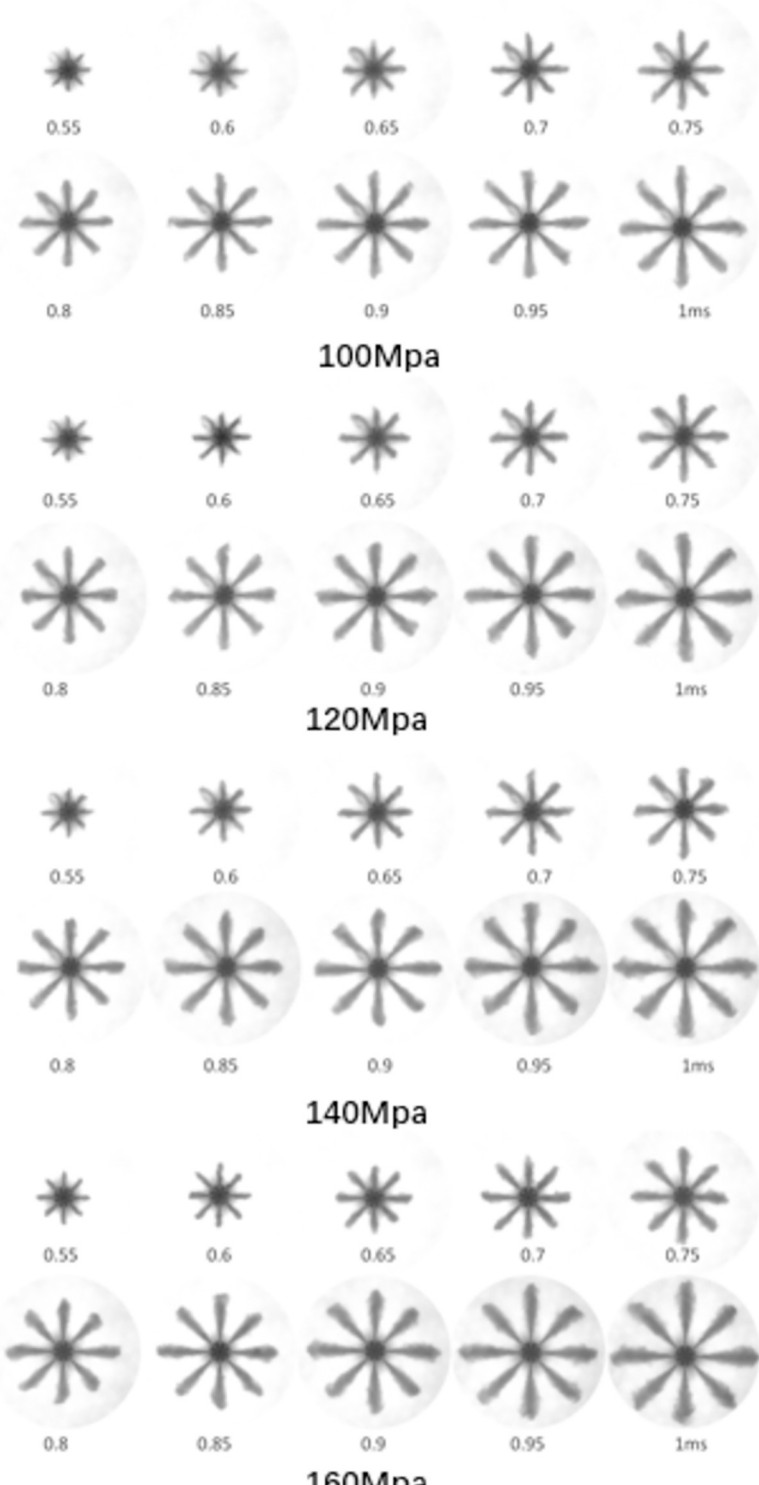

**Fig 8. Spray morphology (time in ms ASOI) at various $P_{in}$ and fixed $P_{out}$ of 5MPa.**

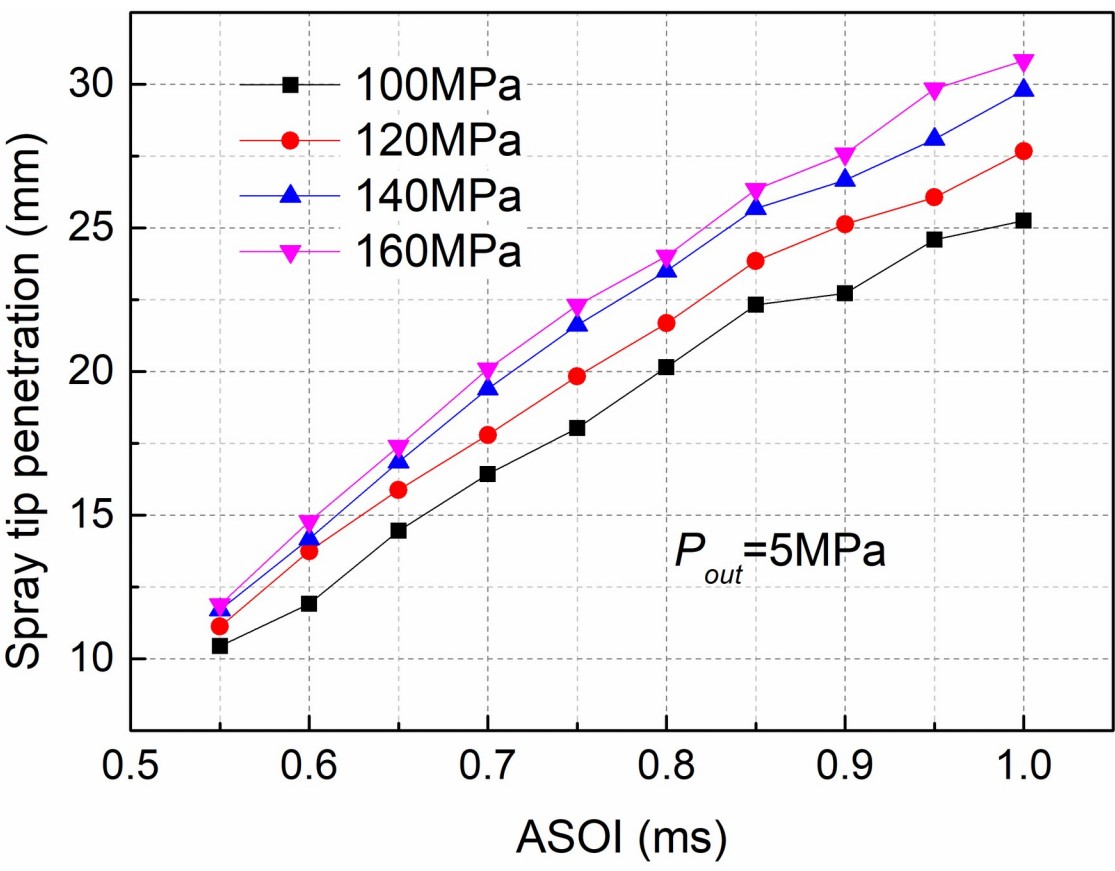

**Fig 9. Spray tip penetration at various $P_{in}$ and fixed $P_{out}$ of 5MPa.**

$P_{out}$ of 5MPa. These images were captured to study macroscopic spray characteristics. The macro spray images provided qualitative parameters for different injection pressures. In addition, the macro spray characteristics, such as the spray tip penetration and the spray cone angle at various injection pressures, were analyzed.

Fig 9 presents the development of spray penetration at various injection pressures and ambient pressure of 5MPa. As shown in Fig 9, when comparing the penetration of various injection pressures, the results indicated that the penetration level at an injection pressure of 160MPa was the highest, and with increasing $P_{in}$, the spray tip penetration increased. The results showed that increasing the $P_{in}$ can increase the spray tip penetration. This difference was due to the high $P_{in}$ increasing the spray momentum. Moreover, as previously stated in Fig 7, cavitation becomes more intense at an injection pressure of 160MPa. However, the intensity of cavitation inside the nozzle was weaker under the condition of low $P_{in}$ (100MPa). Thus, the generation of cavitation reduced the actual orifice circulation area, and bubbles adheringon the inner wall of the nozzle could decrease the friction between the nozzle orifice wall and the fuel, which could improve the development of spray tip penetration.

Fig 10 shows the spray cone angle at various $P_{in}$ and $P_{out}$ of 5MPa. When comparing the four injection pressures, the effect of the $P_{in}$ on the spray cone angle was not evident. However, a large difference was particularly evident at the early stage of the spray. For instance, the spray cone angle for high $P_{in}$ (160MPa) was 21.7% higher than that for low $P_{in}$ (100MPa) at an injection time of 0.55ms. As stated in a previous study shown in Fig 7, the cavitation intensity at high $P_{in}$ (160MPa) is more intense than that at low $P_{in}$ (100MPa), which indicates that

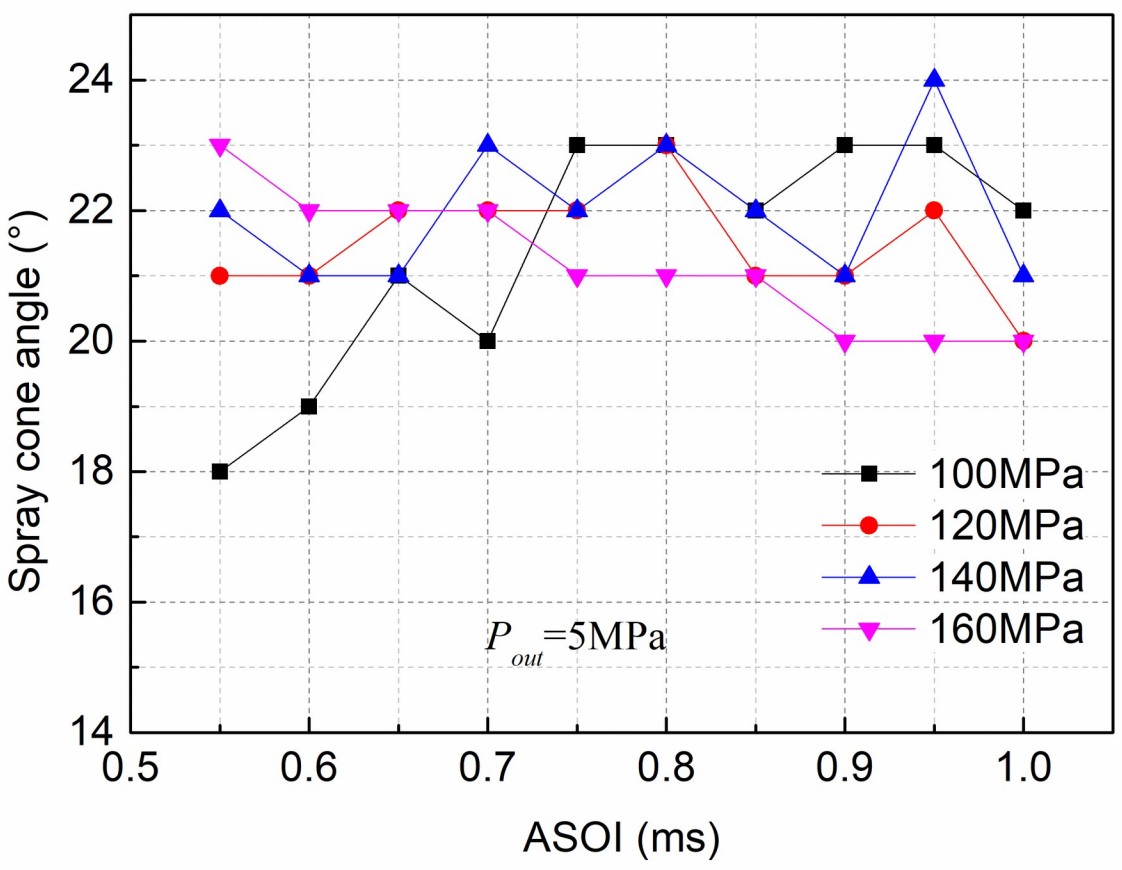

**Fig 10. Spray cone angle at various $P_{in}$ and fixed $P_{out}$ of 5MPa.**

cavitation produces an increase in the spray cone angle, mainly due to broken cavitation bubbles, which induce many micro fluid jets and increase the radial velocity of the jet, thus increasing the spray cone angle, especially during the primary spray process [27]. Additionally, the broken cavitation bubbles can advance the primary breakup process and improve the spray quality.

### Effects of $P_{out}$ on internal flow and spray characteristics

**Simulation results of orifice exit flow parameters.**  Fig 11 represents the mass flow rates and discharge coefficients at various $P_{out}$ when the $P_{in}$ is fixed at 120MPa. Fig 11 shows that the mass flow rate remains the same with increasing $P_{out}$. This is due to choked flow occurring, and the mass flow rate is always choked, remaining unchanged regardless of the $P_{out}$ [28]. In addition, the discharge coefficient increases with increasing $P_{out}$. This means that the flow performance of the orifice is slightly increased at a high backpressure, mainly due to the decrease in cavitation intensity in a nozzle with high $P_{out}$, which can suppress the cavitation disturbance, thus improving the flow performance of the orifice.

**Simulation results of vapor volume fraction distribution.**  Fig 12 shows the vapor volume fraction under different $P_{out}$ (1MPa, 2MPa, 3MPa, and 4MPa)when the $P_{in}$ is fixed at 120MPa. For the vapor phase fraction contour, the domain is colored red where cavitation is generated. The results indicate that the cavitation domain is generated near the inlet corner of the nozzle, and the cavitation becomes more intense with decreasing $P_{out}$. In addition, it can

**Fig 11. Mass flow rate and discharge coefficient at various $P_{out}$ and fixed $P_{in}$ of 120MPa.**

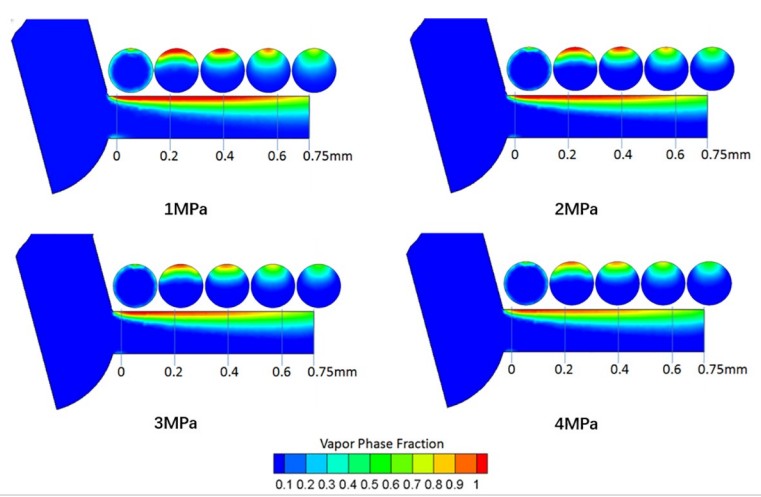

**Fig 12. Contour of vapor volume fraction against different $P_{out}$ at various positions and fixed $P_{in}$ of 120MPa.**

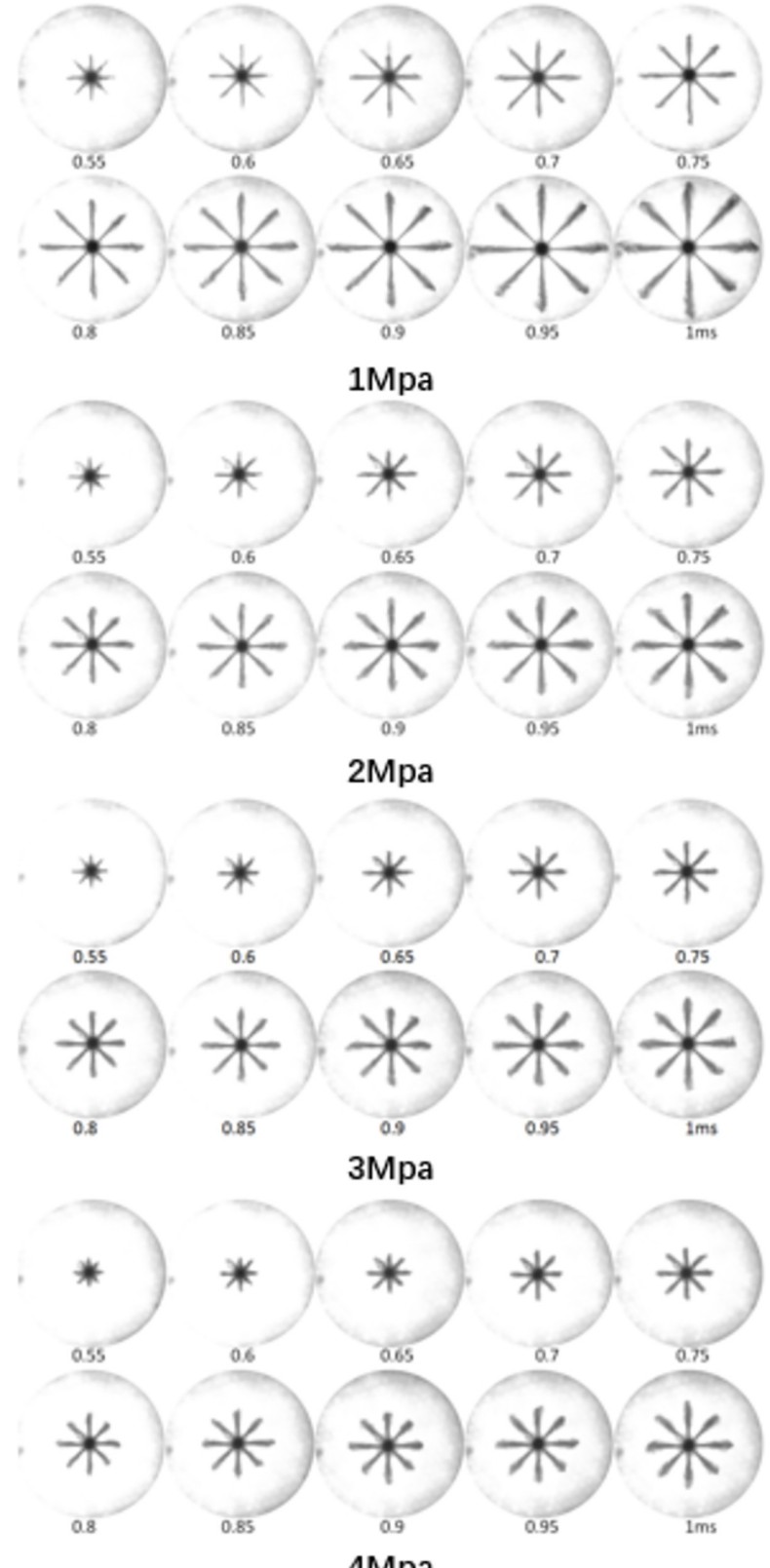

**Fig 13. Spray morphology (time in ms ASOI) under different $P_{out}$ and fixed $P_{in}$ of 120MPa.**

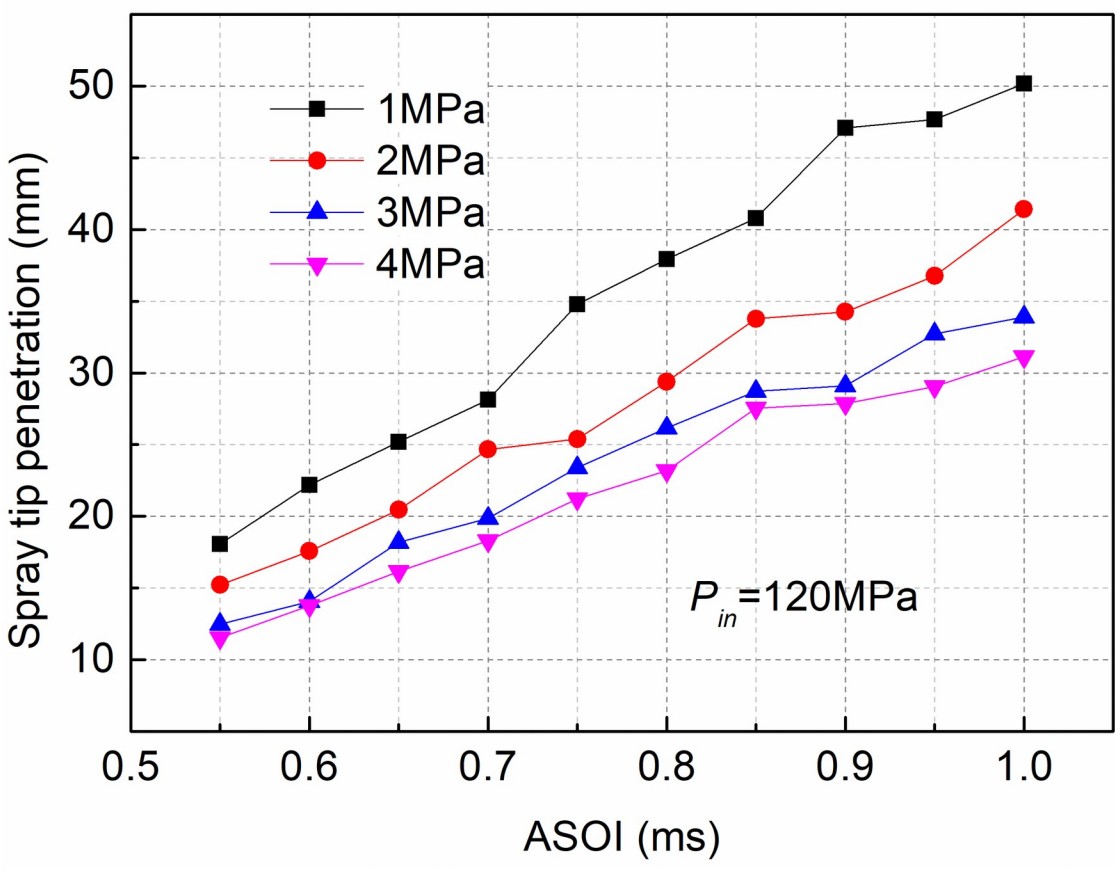

**Fig 14. Spray tip penetration at various $P_{out}$ and fixed $P_{in}$ of 120MPa.**

be seen from the red domain inside the nozzle that the red domain extends to the nozzle exit at the low $P_{out}$ of 1MPa, but for the high $P_{out}$ of 4MPa, this phenomenon did not occur, which means that decreasing the $P_{out}$ can enhance the formation of cavitation. Moreover, the cavitation bubbles break at the nozzle exit, which can increase the cavitation disturbance in the diesel nozzle and thus decrease the flow performance of the nozzle. This is also the reason for the lowest discharge coefficient at the low $P_{out}$ of 1MPa.

**Experimental results of spray characteristics.** This study shows the spray morphology obtained from the experimental spray method at different $P_{out}$. Fig 13 shows the evolution of the spray morphology at different $P_{out}$ of 1MPa, 2MPa, 3MPa, and 4MPa with the $P_{in}$ fixed at 120MPa. These images were captured to analyze macroscopic spray characteristics. The macro spray images provide detailed spray parameters for different $P_{out}$. In addition, the macro spray characteristics at various $P_{in}$ were compared.

Fig 14 illustrates the development of the spray penetration at various back pressures and the $P_{in}$ was fixed at 120MPa. As shown in Fig 14, when comparing the penetration of various back pressures, the results showed that the penetration for the low $P_{out}$ of 1MPawasthe highest, and with increasing $P_{out}$, the spray tip penetration decreased. It can be concluded that increasing the $P_{out}$ can decrease the spray tip penetration. This difference is due to the high $P_{out}$ decreasing the jet velocity and the spray momentum. In turn, the spray penetration was reduced. Moreover, as previously stated in Fig 12, cavitation became more intense under low ambient pressure (1MPa) than under high $P_{out}$ (4MPa). Thus, the generation of cavitation reduced the orifice actual

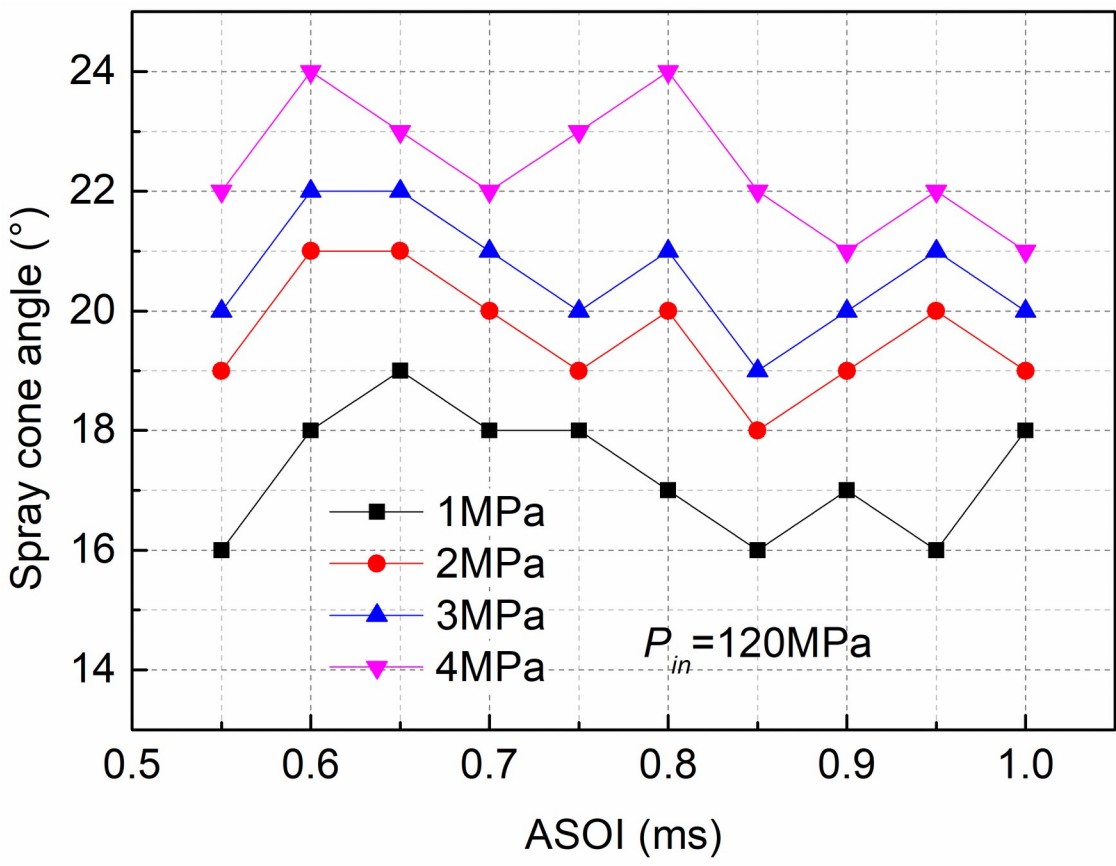

**Fig 15. Spray cone angle at various $P_{out}$ and fixed $P_{in}$ of 120MPa.**

circulation area, and cavitation bubbles adhering to the inner wall of the nozzle could decrease the friction between the nozzle wall and the fuel, which could increase spray tip penetration.

Fig 15 shows the spray cone angle at different $P_{out}$ and the $P_{in}$ was fixed at 120MPa. The results indicated that the spray cone angle was wider with high $P_{out}$ (4MPa) than with low $P_{out}$ (1MPa). Moreover, the large difference in spray cone angle occurred at the early stage of spraying; in particular, the spray cone angle for the $P_{out}$ of 4 MPa was 27.3% higher than that for the low $P_{out}$ (1MPa) at an injection time of 0.55ms. The cavitation bubbles at the nozzle exit were more likely to break under high $P_{out}$ and overcome the surface tension, which can induce many micro fluid jets and increase the radial velocity of the jet, thus increasing the spray cone angle, especially during the initial spraying process [29]. Additionally, a larger ambient pressure can enhance the interaction between the spray and the air, which can greatly increase the quality of the spray.

## Conclusion

In the present work, numerical results of inner flow patterns were studied and combined with the results of macroscopic spray characteristics obtained from experiment methods for various $P_{in}$ and $P_{out}$. On the basis of our research, the following conclusions are made:

(1)An increase in the $P_{in}$ is likely to improve the mass flow rate and decrease the discharge coefficient, thus lowering the flow performance of the nozzle. Moreover, the vapor phase

volume fraction is enhanced with the increase in $P_{in}$, which indicates that a high $P_{in}$ can improve the generation of cavitation inside the nozzle.

(2)With increasing $P_{in}$, the spray tip penetration increased. The effect of $P_{in}$ on the spray cone angle was not evident. However, a large difference was noted during the initial spraying process, especially at the start of injection at0.55 ms, where the spray cone angle for high $P_{in}$ (160MPa) was 21.7% higher than that for low $P_{in}$ (100MPa).

(3) With increasing $P_{out}$, the mass flow rate at the nozzle exit remains the same due to choked flow. However, the discharge coefficient increases with increasing $P_{out}$. In addition, decreasing the $P_{out}$ can enhance the formation of cavitation.

(4)Increasing the $P_{out}$ can decrease the spray tip penetration, while the spray cone angle is wider at a high $P_{out}$ of 4MPa compared to a low $P_{out}$ of 1MPa, especially during the initial spray. Additionally, a larger $P_{out}$ can promote the interaction of the spray and the air, which can greatly increase the spray quality.

## Author Contributions

**Conceptualization:** Hua Xia.

**Data curation:** Hua Xia.

**Formal analysis:** Hua Xia.

**Funding acquisition:** Hua Xia.

**Investigation:** Hua Xia.

**Methodology:** Hua Xia.

**Project administration:** Hua Xia.

**Resources:** Hua Xia.

**Software:** Hua Xia.

**Supervision:** Hua Xia.

**Validation:** Hua Xia.

**Visualization:** Hua Xia.

**Writing – original draft:** Hua Xia.

**Writing – review & editing:** Hua Xia.

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
