## [Decision Letter · Decision Letter 0]

19 Apr 2021

PONE-D-20-28234

Research on the Internal Flow and Macroscopic Characteristics of Diesel Fuel Injection Progress

PLOS ONE

Dear Dr. Xia,

Thank you for submitting your manuscript to PLOS ONE. After careful consideration, we feel that it has merit but does not fully meet PLOS ONE’s publication criteria as it currently stands. Therefore, we invite you to submit a revised version of the manuscript that addresses the points raised during the review process.

We look forward to receiving your revised manuscript.

Kind regards,

Hongbing Ding, Ph.D.

Academic Editor

PLOS ONE

Journal Requirements:

2. Please amend either the title on the online submission form (via Edit Submission) or the title in the manuscript so that they are identical.

3. Please ensure that you refer to Figure 5 in your text as, if accepted, production will need this reference to link the reader to the figure.

Additional Editor Comments:

Thank you for submitting your manuscript to PLOS ONE. I have completed the review of your manuscript and a summary is appended below. The reviewers recommend reconsideration of your paper following major revision. I invite you to resubmit your manuscript after addressing all reviewer comments.

When resubmitting your manuscript, please carefully consider all issues mentioned in the reviewers' comments, outline every change made point by point, and provide suitable rebuttals for any comments not addressed.

Reviewers' comments:

Reviewer's Responses to Questions

**Comments to the Author**

1. Is the manuscript technically sound, and do the data support the conclusions?

Reviewer #1: Yes

2. Has the statistical analysis been performed appropriately and rigorously? 

Reviewer #1: Yes

3. Have the authors made all data underlying the findings in their manuscript fully available?

Reviewer #1: Yes

4. Is the manuscript presented in an intelligible fashion and written in standard English?

Reviewer #1: Yes

5. Review Comments to the Author

Reviewer #1: Paper title: Research on the Internal Flow and Macroscopic Characteristics of a Diesel Fuel Injection Process

The paper studied the internal flow and the macroscopic characteristics of a diesel fuel injection process. The topic is interesting, the results are discussed in detail. However, there are some flaws to be revised, also the organization is not proper. Here are the detailed comments:

1. The organization of paper needs to be adjusted. The authors use both CFD and experimental methods to study the topic, thus the research object (nozzle) as well as the studied conditions (in CFD, it is expressed as boundary conditions, in experiment, it is the experimental conditions, in either way, the detailed conditions should be the same). Consequently, the nozzle geometry (Fig. 1 and Table 1) and the studied conditions (Table 2) should not be placed in CFD part.

2. As I have mentioned, the nozzle structure and the studied conditions should be placed in another part. In addition, the CFD results and the experiment results should be clearly stated, rather than be confused with each other.

3. As the most results are obtained from experiment, only the “Internal flow” part uses the CFD results (if I understand correctly), I suggest the “Description of the CFD method” merges with the “Internal flow”. In this case, the readers can clearly know the CFD study is conducted for the internal flow analysis, not the spray analysis. In this case, the detailed CFD setup, eg., govern equations can choose not be listed. If the authors believe the CFD plays as important role, then more detailed CFD setup, eg., govern equations and CFD results validation, needs to be listed.

4. The authors separated the results by the effect of the injection and ambient pressures, in my opinion, it is not a good choice. The injection and ambient pressures work together to effect the orifice discharge coefficient and the spray characteristics. Therefore, the authors are suggested to discuss these two pressures together, especially the discharge coefficient, it is a function of the differential pressure. In addition, the subtitle “Internal flow” is not clear, maybe the “Volume fraction distribution” is better.

5. The results should be discussed more depth combined with the spray process and breakup mechanism. In addition, the important conclusions and explanations needs to be supported by the detailed results or the literature references, eg., in the beginning of Page 17 “which indicates that cavitation produces an increase in the spray cone angle, mainly due to broken cavitation bubbles, which induce many micro fluid jets and increase the radial velocity of the jet, thus increasing the spray cone angle, especially during the primary spray process”. How the authors draw the conclusion without any broken cavitation bubbles pictures or any references? There are many sentences not rigorous enough.

6. Fig.1 is not clear to show the detailed structure of the nozzle, the authors should considered the section map.

7. In general, the symbols indicating physical quantity symbols are in italics, and the others are in roman. Please check whether all the subscripts and symbols in this paper are used correctly.

6. PLOS authors have the option to publish the peer review history of their article (what does this mean?). If published, this will include your full peer review and any attached files.

Reviewer #1: No

---

## [Author Response · Author response to Decision Letter 0]

25 May 2021

Response to Reviewers

Dear Editor-in-chief:

We have considered the suggestions of the reviewers carefully and re-submitted the manuscript according to your kind advice. Thank you for your letter and for the reviewers’ comments concerning our manuscript entitled “Research on the Internal Flow and Macroscopic Characteristics of a Diesel Fuel Injection Process”. Those comments are all valuable and very helpful for revising and improving our paper, as well as the important guiding significance to our researches. We have studied comments carefully and have made correction which we hope meet with approval. The corrections in the paper and the responds to the reviewer’s comments are as following:

Editor 

2. Please amend either the title on the online submission form (via Edit Submission) or the title in the manuscript so that they are identical.

3. Please ensure that you refer to Figure 5 in your text as, if accepted, production will need this reference to link the reader to the figure.

Response: Thank you for your comments. I have revised the article as required.

Reviewer #1: Paper title: Research on the Internal Flow and Macroscopic Characteristics of a Diesel Fuel Injection Process

The paper studied the internal flow and the macroscopic characteristics of a diesel fuel injection process. The topic is interesting, the results are discussed in detail. However, there are some flaws to be revised, also the organization is not proper. Here are the detailed comments:

1. The organization of paper needs to be adjusted. The authors use both CFD and experimental methods to study the topic, thus the research object (nozzle) as well as the studied conditions (in CFD, it is expressed as boundary conditions, in experiment, it is the experimental conditions, in either way, the detailed conditions should be the same). Consequently, the nozzle geometry (Fig. 1 and Table 1) and the studied conditions (Table 2) should not be placed in CFD part.

Response: Thank you for your comments. I adjusted the position of the nozzle geometry and the studied conditions in the paper. For details, please refer to the revision of Chapter 2 and Chapter 3 in the paper.

2. As I have mentioned, the nozzle structure and the studied conditions should be placed in another part. In addition, the CFD results and the experiment results should be clearly stated, rather than be confused with each other.

Response：Thank you for your comments. The result of inner flow is obtained through CFD simulation. The macro spray characteristics are obtained through experiments. I have stated these clearly in our revised manuscript.

3. As the most results are obtained from experiment, only the “Internal flow” part uses the CFD results (if I understand correctly), I suggest the “Description of the CFD method” merges with the “Internal flow”. In this case, the readers can clearly know the CFD study is conducted for the internal flow analysis, not the spray analysis. In this case, the detailed CFD setup, eg., govern equations can choose not be listed. If the authors believe the CFD plays as important role, then more detailed CFD setup, eg., govern equations and CFD results validation, needs to be listed.

Response：Thank you for your comments. Yes, only the internal flow part uses the CFD results, and the most results are obtained from experiment. I Combine "Description of CFD Method" with "Internal Flow" as you suggest. Therefore, the detailed CFD setup and govern equations were not listed in the manuscript. The simulation results and the experiment results were clearly stated in our revised manuscript.

4. The authors separated the results by the effect of the injection and ambient pressures, in my opinion, it is not a good choice. The injection and ambient pressures work together to effect the orifice discharge coefficient and the spray characteristics. Therefore, the authors are suggested to discuss these two pressures together, especially the discharge coefficient, it is a function of the differential pressure. In addition, the subtitle “Internal flow” is not clear, maybe the “Volume fraction distribution” is better.

Response：Thank you for your comments. Firstly, the change of injection pressure and ambient back pressure have different effects on the cavitation and the downstream spray behaviors. In the case of high-pressure injection, as the injection pressure increases, the flow characteristics inside the nozzle will be affected. In the case of a certain fuel injection pressure, the change of the environmental back pressure has a great influence on the fuel spray. In order to explore the influence of the fuel injection pressure and the environmental back pressure on the nozzle internal flow characteristics and the fuel spray characteristics. That's why they were studied separately from each other. Secondary, the discharge coefficient is a function of the differential pressure, when we study the effectsof injection pressure or the back pressure on Cd, we always fix a constant pressure value, and only change the injection pressure or the back pressure.The effect is the same as changing the differential pressure. Meanwhile, the internal flow has been changed with vapor volume fraction distribution.

5. The results should be discussed more depth combined with the spray process and breakup mechanism. In addition, the important conclusions and explanations needs to be supported by the detailed results or the literature references, eg., in the beginning of Page 17 “which indicates that cavitation produces an increase in the spray cone angle, mainly due to broken cavitation bubbles, which induce many micro fluid jets and increase the radial velocity of the jet, thus increasing the spray cone angle, especially during the primary spray process”. How the authors draw the conclusion without any broken cavitation bubbles pictures or any references? There are many sentences not rigorous enough.

Response：Thank you for your comments. Based on your opinion, I have carefully revised the results analysis, interpretation and relevant conclusions in the full text.

Combining with the questions you raised on page 17, the spraying process and the crushing mechanism, the results in the paper are analyzed and compared with the conclusions of the references, and the references are marked in the corresponding positions of the article. See the red part in the article. 

6. Fig.1 is not clear to show the detailed structure of the nozzle, the authors should considered the section map.

Response：Thank you for your comments. Figure 1 is a cross-sectional view of the nozzle needle according to the parts drawing, and the schematic diagram of the nozzle structure has been modified. The enlarged schematic diagram of the nozzle hole has been added and the relevant geometric parameters have been marked. And add Table 1 to supplement the nozzle structure parameters. The relevant content has been revised in the paper and marked in red.

7. In general, the symbols indicating physical quantity symbols are in italics, and the others are in roman. Please check whether all the subscripts and symbols in this paper are used correctly.

Response：Thank you for your comments. Sorry for our unclear writing, we have check the symbols and revised in our revised manuscript.

We have studied reviewer’s comments carefully and have made revision which marked in red in the paper. We have tried our best to revise our manuscript according to the comments, and We would like to submit for your kind consideration.We would like to express our great appreciation to you and reviewers for comments on our paper. Looking forward to hearing from you.

Thank you and best regards.

Yours sincerely

Corresponding author 

Name :Hua Xia

E-mail: 250930192@qq.com

---

## [Decision Letter · Decision Letter 1]

7 Jul 2021

PONE-D-20-28234R1

Research on the Internal Flow and Macroscopic Characteristics of Diesel Fuel Injection Progress

PLOS ONE

Dear Dr. Xia,

Thank you for submitting your manuscript to PLOS ONE. After careful consideration, we feel that it has merit but does not fully meet PLOS ONE’s publication criteria as it currently stands. Therefore, we invite you to submit a revised version of the manuscript that addresses the points raised during the review process.

We look forward to receiving your revised manuscript.

Kind regards,

Hongbing Ding, Ph.D.

Academic Editor

PLOS ONE

Additional Editor Comments (if provided):

Thank you for submitting your manuscript to PLOS ONE. The reviewers recommend reconsideration of your paper following minor revision. I invite you to resubmit your manuscript after addressing all reviewer comments.

Journal Requirements:

Reviewers' comments:

Reviewer's Responses to Questions

**Comments to the Author**

1. If the authors have adequately addressed your comments raised in a previous round of review and you feel that this manuscript is now acceptable for publication, you may indicate that here to bypass the “Comments to the Author” section, enter your conflict of interest statement in the “Confidential to Editor” section, and submit your "Accept" recommendation.

Reviewer #1: (No Response)

2. Is the manuscript technically sound, and do the data support the conclusions?

Reviewer #1: Yes

3. Has the statistical analysis been performed appropriately and rigorously? 

Reviewer #1: Yes

4. Have the authors made all data underlying the findings in their manuscript fully available?

Reviewer #1: Yes

5. Is the manuscript presented in an intelligible fashion and written in standard English?

Reviewer #1: Yes

6. Review Comments to the Author

Reviewer #1: The revised paper is largely improved compared with the original version, and the most problems I mentioned have been addressed. I have some comments to further improve the quality of the paper.

1. The subsection title “The boundary conditions” is not proper. As I have mentioned, the boundary conditions are the exclusive expression in CFD simulation, it includes the boundary setup in CFD. Maybe “The groups of operational conditions” is better. In addition, the injection pressure and ambient pressure are strongly suggested to be named with the variable symbol, for example, Pin and P0, as well as in the whole text and the figures. It is more clear and readable.

2. The variable symbols are not rigorous, please check the paper thoroughly to ensure all the subscript and symbol are identical. For example, K-factor in equation (1), it is K or k？In equation (2), P should be italic, the symbol “Cd” in Fig. 6 should be identical with it in equation (2).

3. The captions of the figure is not clear, they should be self-evidence. The detailed operational conditions of the results should be given clearly. For example, what is the ambient pressure in Fig.10？The detailed information should be given. In addition, Fig. 7 has typographical error. As I have mentioned, the corresponding variable symbols are suggested to add.

4. The logical relation and relevance between the experimental and numerical results should be more clearer.

5. There are still some grammatical or typographical errors, please check it clearly to avoid it.

7. PLOS authors have the option to publish the peer review history of their article (what does this mean?). If published, this will include your full peer review and any attached files.

Reviewer #1: No

---

## [Author Response · Author response to Decision Letter 1]

23 Jul 2021

Response to Reviewers

Dear Editor and reviewers

Thank you for your view letter and the comments concerning our manuscript entitled ‘Research on the Internal Flow and Macroscopic Characteristics of a Diesel Fuel Injection Process’. Those comments are all very valuable and helpful for revising and improving our paper. We have studied comments carefully and have made correction which we hope meet with approval. The corrections in the paper and the responds to the comments are as following:

Reviewer #1: The revised paper is largely improved compared with the original version, and the most problems I mentioned have been addressed. I have some comments to further improve the quality of the paper.

1. The subsection title “The boundary conditions” is not proper. As I have mentioned, the boundary conditions are the exclusive expression in CFD simulation, it includes the boundary setup in CFD. Maybe “The groups of operational conditions” is better. In addition, the injection pressure and ambient pressure are strongly suggested to be named with the variable symbol, for example, Pin and P0, as well as in the whole text and the figures. It is more clear and readable.

Response: Thank you for your comments. We have revised these in our revised manuscript. is the injection pressure, is the ambient pressure. We have changed the original title to “The groups of operational conditions”.

2. The variable symbols are not rigorous, please check the paper thoroughly to ensure all the subscript and symbol are identical. For example, K-factor in equation (1), it is K or k？In equation (2), P should be italic, the symbol “Cd” in Fig. 6 should be identical with it in equation (2).

Response: Thank you for your comments. Sorry for our unclear writing, it is K; We have revised these in our revised manuscript. We have checked the full text to prevent similar errors.

3. The captions of the figure is not clear, they should be self-evidence. The detailed operational conditions of the results should be given clearly. For example, what is the ambient pressure in Fig.10？The detailed information should be given. In addition, Fig. 7 has typographical error. As I have mentioned, the corresponding variable symbols are suggested to add.

Response: Thank you for your comments. The ambient pressure in Fig.10 is 5MPa, we have added in the manuscript. We have modified the titles of several figures to make them more accurate.

4. The logical relation and relevance between the experimental and numerical results should be more clearer.

Response: Thank you for your comments. The computational results of inner flow in nozzle orifice are used to analyze the mechanism and reason of the difference between the experimental results of spray characteristics. This part was explained in the part of spray macroscopic characteristics.

5. There are still some grammatical or typographical errors, please check it clearly to avoid it.

Response: Thank you for your comments. We have revised the grammatical or typographical errors in our revised manuscript ensure that readers can clearly understand the paper.

We have studied reviewer’s comments carefully and we have tried our best to revise our manuscript according to the comments, and the words in red font were the modified content in the revised manuscript. We would like to submit for your kind consideration. We would like to express our great appreciation to you for comments on our paper. Looking forward to hearing from you.

Thank you and best regards.

Yours sincerely

Corresponding author 

Name: Hua Xia

E-mail: 250930192@qq.com

---

## [Editor Report · Decision Letter 2]

27 Jul 2021

Research on the Internal Flow and Macroscopic Characteristics of Diesel Fuel Injection Progress

PONE-D-20-28234R2

Dear Dr. Xia,

We’re pleased to inform you that your manuscript has been judged scientifically suitable for publication and will be formally accepted for publication once it meets all outstanding technical requirements.

Kind regards,

Hongbing Ding, Ph.D.

Academic Editor

PLOS ONE

Additional Editor Comments (optional):

The authors have done a good job in revising the manuscript. Now it can be accepted for publication in PLOS ONE.
---

## [Editor Report · Acceptance letter]

17 Sep 2021

PONE-D-20-28234R2 

Research on the Internal Flow and Macroscopic Characteristics of a Diesel Fuel Injection Process 

Dear Dr. Xia:

I'm pleased to inform you that your manuscript has been deemed suitable for publication in PLOS ONE. Congratulations! Your manuscript is now with our production department. 

Kind regards, 

on behalf of

Professor Hongbing Ding 

Academic Editor

PLOS ONE